# Prognostic factors for favorable outcomes after veno-venous extracorporeal membrane oxygenation in critical care patients with COVID-19

**Bärbel Kieninger[1], Magdalena Kilger[2], Maik Foltan[3], Michael Gruber®[2], Dirk Lunz[2], Thomas Dienemann[4], Stephan Schmid[5], Bernhard Graf[2], Clemens Wiest[6], Matthias Lubnow[6], Thomas Müller[6], Bernd Salzberger[1], Wulf Schneider-Brachert[1], Martin Kieninger®[2]***

1 Department of Infection Prevention and Infectious Diseases, University Medical Center Regensburg, Regensburg, Germany, 2 Department of Anesthesiology, University Medical Center Regensburg, Regensburg, Germany, 3 Department of Cardiac, Thoracic and Cardiovascular Surgery, University Medical Center Regensburg, Regensburg, Germany, 4 Department of Surgery, University Medical Center Regensburg, Regensburg, Germany, 5 Department of Internal Medicine I, University Medical Center Regensburg, Regensburg, Germany, 6 Department of Internal Medicine II, University Medical Center Regensburg, Regensburg, Germany

* martin.kieninger@ukr.de

**Data Availability Statement:** All relevant data are within the paper and its Supporting Information files.

## Abstract

### Background

Patients with COVID-19 and severe acute respiratory failure may require veno-venous extracorporeal membrane oxygenation (VV ECMO). Yet, this procedure is resource-intensive and high mortality rates have been reported. Thus, predictors for identifying patients who will benefit from VV ECMO would be helpful.

### Methods

This retrospective study included 129 patients with COVID-19 and severe acute respiratory failure, who had received VV ECMO at the University Medical Center Regensburg, Germany, between 1 March 2020 and 31 December 2021. Patient-specific factors and relevant intensive-care parameters at the time of the decision to start VV ECMO were investigated regarding their value as predictors of patient survival. In addition, the intensive-care course of the first 10 days of VV ECMO was compared between survivors and patients who had died in the intensive care unit.

### Results

The most important parameters for predicting outcome were patient age and platelet count, which differed significantly between survivors and non-survivors (age: 52.6±8.1 vs. 57.4 ±10.1 years, p<0.001; platelet count before VV ECMO: 321.3±132.2 vs. 262.0±121.0 /nL, p = 0.006; average on day 10: 199.2±88.0 vs. 147.1±57.9 /nL, p = 0.002). A linear regression model derived from parameters collected before the start of VV ECMO only included age and platelet count. Patients were divided into two groups by using receiver operating

**Funding:** The authors received no specific funding for this work.

**Competing interests:** Please take note of the following statement regarding Martin Kieninger's service as Academic Editor for PLOS One: This does not alter our adherence to PLOS ONE policies on sharing data and materials.

characteristics (ROC) analysis: group 1: 78% of patients, mortality 26%; group 2: 22% of patients, mortality 75%. A second linear regression model included average blood pH, minimum paO2, and average pump flow on day 10 of VV ECMO in addition to age and platelet count. The ROC curve resulted in two cut-off values and thus in three groups: group 1: 25% of patients, mortality 93%; group 2: 45% of patients, mortality 31%; group 3: 30% of patients, mortality 0%.

## Introduction

Up to 5% of patients with COVID-19 develop severe respiratory failure [1]. Patients with particularly severe disease may require veno-venous extracorporeal membrane oxygenation (VV ECMO) to maintain adequate pulmonary gas exchange. Yet, the mortality rate of patients treated with VV ECMO in Germany was strikingly high by international comparison [2–4]. Furthermore, VV ECMO is a resource-intensive therapy against the backdrop of limited resources and is ideally applied at specialized centers. Therefore, it would be desirable to have cut-off values for certain parameters to identify patients who are most likely to benefit from VV ECMO. Such predictive values would be even more important because of the relatively long treatment duration of 9 to 20 days of patients with COVID-19 who require VV ECMO because of acute respiratory failure [3–5].

More than 150 patients with COVID-19 have been treated with VV ECMO at the University Medical Center Regensburg in Germany so far, resulting in a very large data pool for this particular patient population. The aim of the present study was to use this data pool to determine cut-off values for baseline parameters at the time of the decision to start VV ECMO, indicating that the patient will benefit from this therapy. In addition, we compared the course over the first 10 days of VV ECMO between survivors and patients who had died in the intensive care unit (ICU).

## Material and methods

### Aim of the study

First, we retrospectively investigated the data sets of patients with COVID-19 who had received VV EMCO because of respiratory failure with regard to the presence of any patient-specific factors or cut-off values for parameters relevant to intensive care at the time of the decision to start VV ECMO that are associated with good outcome. Additionally, we aimed to identify any cut-off values for these parameters associated with good outcome over time.

### Ethics approval and consent to participate

The study was approved by and conducted according to the guidelines of the Ethics Committee of the University of Regensburg (approval number 20-1790-104). In accordance with European law, consent to participate was not required because of the retrospective study design and the use of anonymized patient data. All data were anonymized prior to analysis.

### Patients and settings

We considered all patients with COVID-19 and acute respiratory failure who had been treated with VV ECMO at the University Medical Center Regensburg, Germany, between 1 March 2020 and 31 December 2021. VV EMCO had to be initiated at the University Medical Center Regensburg within 14 days after admission to the ICU or at an external hospital not longer

than 24 hours before the transfer to the University Medical Center Regensburg. Patients younger than 18 years and pregnant women were excluded.

## Data collection

We examined patient-specific factors as well as parameters relevant to intensive care immediately before the start of VV ECMO and on day 1, 3, 5, and 10 after the start of VV ECMO. These factors and parameters were correlated with outcome for each individual patient (patient died in the intensive care unit vs. patient was discharged alive from the intensive care unit). Data collection was terminated at the end of VV ECMO if this timepoint preceded the end of the observation period.

The complete list of all parameters examined is provided in 'S1 Table.' Complete data sets consisted of 117 values per patient. Data were extracted from the data management systems of the ICUs (MetaVisionSuite®, version V6.9.0.23, iMDsoft®, Tel Aviv, Israel; SAP® Enterprise resource planning, version 6.0 EHP7 SP21, SAP SE, Walldorf, German; SWISSLAB® Laborinformationssysteme, version 2.18.3.00, NEXUS SWISSLAB GmbH, Berlin, Germany).

## Statistical analysis

Statistical analysis was conducted using IBM SPSS Statistics[TM] 28 (IBM, Armonk, USA). Statistical tests were two-sided, and the level of significance was set to $p < 0.050$ (termed 'significant') and to $p < 0.010$ (termed 'highly significant'). Categorical parameters are presented as absolute and relative frequencies. Data of survivors and non-survivors were compared with the Chi-square test of independence. Continuous data are shown as mean ± standard deviation (SD) as well as minimum and maximum and the mean difference (MD) is provided. Differences between survivors and non-survivors were assessed with the Mann-Whitney-U test.

Binary logistic regression models with the predictor variable 'outcome' were calculated with the parameters that showed significant differences between survivors and non-survivors for both, the baseline parameters at the time of the decision to start VV ECMO and the parameters derived 10 days after the start of VV ECMO. To determine the best possible set of parameters derived 10 days after the start of VV ECMO, the number of included parameters was limited to a maximum of five. The composition of the set was varied until a consistent set of parameters was found with only significant parameters for the regression model. Multicollinearity between these parameters was assessed by using Pearson's correlation coefficient. Odds ratios and 95%-confidence intervals, p values, and regression coefficients are reported for all logistic regression models. Receiver operating characteristics (ROC) curves were used for the evaluation of the model. In addition, ROC curves were used to determine appropriate cut-off values for the model to divide patients into groups. Mortality was calculated for each group.

## Results

Between 1 March 2020 and 31 December 2021, 356 patients with COVID-19 had been treated at one of the ICUs of the University Medical Center Regensburg. Of these 356 patients, 129 (36.2%) were treated with VV ECMO and also met the above inclusion criteria (S1 Fig). The underlying data of all included patients and technical data on ECMO therapy are presented in 'S1 and S2 Appendices'.

### Baseline and demographic data

81 patients had been discharged from the ICU ('survivors'), and 48 (37.2%) patients had died in the ICU ('non-survivors'). At the start of VV ECMO, survivors had been significantly

younger than non-survivors (52.6±8.1 vs. 57.4±10.1 years, p<0.001). Most patients were men (101 men vs. 28 women, 78.3%). The group of non-survivors consisted of 40 (83.3%) male and 8 (16.7%) female patients (p = 0.378). Body mass index (BMI) did not differ significantly between survivors and non-survivors (31.3±6.6 kg/m$^2$ vs. 30.1±5.4 kg/m$^2$, p = 0.618). Survivors and non-survivors did not significantly differ with regard to the presence of cardiovascular (p = 0.584), pulmonary (p = 1.000), renal (p = 1.000), endocrinological (p = 0.305), gastrointestinal (p = 1.000), metabolic (p = 0.770), dermatological (p = 0.668), neurological (p = 1.000), gynecological or urological (p = 0.130), malignant (p = 0.142) diseases, obesity (p = 0.855), consumption of noxious substances (p = 1.000), infections (p = 1.000), immunosuppression (p = 0.102), or other diseases. Only orthopedic diseases were less common among survivors (8, 9.9%) than non-survivors (14, 29.2%, p = 0.007). Acute kidney injury before the start of VV ECMO was present in only 1 (1.2%) survivor but in 5 (10.4%) non-survivors (p = 0.025).

The baseline values for survivors and non-survivors for the parameters recorded before the start of VV ECMO therapy are summarized in Table 1.

**Table 1. Baseline parameters before the start of veno-venous extracorporeal membrane oxygenation (VV ECMO).**

| | Survivors | | | | Non-survivors | | | | 95%-CI of MD | | | p |
|---|---|---|---|---|---|---|---|---|---|---|---|---|
| | mean | SD | min | max | mean | SD | min | max | mean | min | max | |
| pH | 7.30 | 0.10 | 7.02 | 7.51 | 7.27 | 0.13 | 6.92 | 7.49 | 0.03 | -0.01 | 0.08 | 1.771E-01 |
| HCO$_3^-$ | 27.8 | 5.7 | 19.0 | 46.2 | 29.2 | 6.9 | 19.0 | 42.0 | -1.4 | -3.9 | 1.0 | 3.114E-01 |
| BE | 2.8 | 5.3 | -6.1 | 18.6 | 4.1 | 5.9 | -7.0 | 16.0 | -1.4 | -3.4 | 0.7 | 2.735E-01 |
| paO$_2$ | 67.6 | 16.5 | 19.6 | 106.0 | 72.7 | 39.4 | 33.0 | 313.0 | -5.0 | -17.0 | 6.9 | 9.024E-01 |
| paCO$_2$ | 62.9 | 16.9 | 38.5 | 140.0 | 72.3 | 24.0 | 38.7 | 162.0 | -9.4 | -17.2 | -1.5 | 2.181E-02 |
| Hb | 11.3 | 2.0 | 8.2 | 17.3 | 10.8 | 2.0 | 7.3 | 14.2 | 0.6 | -0.2 | 1.3 | 1.999E-01 |
| Lactate | 14.3 | 6.4 | 5.0 | 35.0 | 14.9 | 9.1 | 5.0 | 52.0 | -0.6 | -3.6 | 2.4 | 7.652E-01 |
| Creatinine | 1.1 | 0.7 | 0.3 | 3.7 | 1.3 | 0.9 | 0.3 | 4.7 | -0.2 | -0.6 | 0.1 | 1.736E-01 |
| Urea | 60.1 | 38.8 | 13.0 | 273.0 | 72.3 | 35.0 | 15.0 | 159.0 | -12.2 | -25.5 | 1.2 | 2.303E-02 |
| AST | 78 | 70 | 18 | 494 | 80 | 96 | 24 | 515 | -2.7 | -35.0 | 29.7 | 5.250E-01 |
| ALT | 65 | 50 | 16 | 250 | 69 | 64 | 10 | 388 | -4.2 | -26.0 | 17.5 | 7.939E-01 |
| INR | 1.1 | 0.3 | 0.9 | 3.4 | 1.3 | 1.2 | 0.9 | 9.5 | -0.2 | -0.5 | 0.2 | 2.348E-01 |
| LDH | 534 | 386 | 60 | 2905 | 501 | 260 | 175 | 1336 | 33.6 | -80.1 | 147.2 | 9.752E-01 |
| CRP | 146.8 | 135.9 | 1.0 | 547.0 | 160.2 | 119.8 | 1.1 | 506.0 | -13.4 | -59.1 | 32.4 | 3.010E-01 |
| PCT | 1.1 | 1.8 | 0.1 | 11.0 | 2.8 | 6.8 | 0.1 | 39.0 | -1.7 | -3.8 | 0.5 | 1.718E-01 |
| WBC | 13.1 | 8.6 | 2.6 | 68.0 | 13.3 | 6.8 | 3.5 | 37.8 | -0.2 | -3.0 | 2.5 | 4.757E-01 |
| D-Dim | 7.6 | 9.9 | 0.5 | 35.0 | 8.9 | 9.5 | 1.0 | 35.0 | -1.3 | -5.0 | 2.3 | 6.665E-02 |
| Platelets | 321.3 | 132.2 | 107.0 | 744.0 | 262.0 | 121.0 | 67.0 | 603.0 | 59.3 | 13.7 | 104.8 | 6.362E-03 |
| IL6 | 642 | 1687 | 3 | 13324 | 626 | 1380 | 12 | 8140 | 16.2 | -549.0 | 581.4 | 4.319E-01 |
| Nor | 0.8 | 0.9 | 0.0 | 4.6 | 0.8 | 1.0 | 0.0 | 4.2 | 0.0 | -0.4 | 0.3 | 5.807E-01 |
| MAP | 73.7 | 13.2 | 55.0 | 127.0 | 70.4 | 15.1 | 5.0 | 96.0 | 3.3 | -2.0 | 8.6 | 4.251E-01 |
| FiO$_2$ | 95.6 | 13.2 | 0.8 | 100.0 | 95.1 | 11.1 | 60.0 | 100.0 | 0.5 | -3.9 | 4.8 | 7.492E-01 |
| HV | 75.1 | 29.4 | 19.6 | 236.3 | 80.6 | 43.7 | 33.0 | 313.0 | -5.4 | -20.0 | 9.2 | 6.656E-01 |
| PEEP | 14.2 | 3.0 | 4.9 | 21.0 | 14.2 | 2.6 | 5.6 | 19.0 | 0.0 | -1.0 | 1.0 | 6.422E-01 |
| VT | 473.6 | 130.0 | 120.0 | 876.0 | 480.9 | 162.8 | 3.0 | 825.0 | -7.3 | -63.2 | 48.7 | 8.175E-01 |
| Pmean | 21.9 | 3.7 | 13.0 | 33.0 | 21.3 | 3.8 | 12.0 | 35.0 | 0.6 | -0.8 | 2.0 | 5.455E-01 |

Significant differences between survivors and non-survivors are highlighted in orange, highly significant differences in purple. HCO$_3^-$, standard bicarbonate; BE, base excess; paO$_2$, arterial partial pressure of oxygen; paCO$_2$, arterial partial pressure of carbon dioxide; Hb, hemoglobin; AST, aspartate transaminase; ALT, alanine transaminase; INR, International Normalized Ratio; LDH, lactate dehydrogenase; CRP, C-reactive protein; PCT, procalcitonin; WBC, white blood cells; D-Dim, D-dimers; IL6, interleukin 6; MAP, mean arterial pressure; FiO$_2$, fraction of inspired oxygen; HV, oxygenation ratio; PEEP, positive endexpiratory pressure; VT, tidal volume, Pmean, mean airway pressure. The units of the individual parameters can be found in 'S1 Table'

**Table 2. Regression coefficients, significance, and odds ratio with confidence interval for the regression model calculated with the parameters age and initial platelets count (platelets).**

|  | Regression coefficient | p-value | Odds ratio | Confidence interval (95%) for odds ratio |
|---|---|---|---|---|
| Age | 6.360E-02 | 0.007 | 1.066 | 1.017–1.117 |
| Platelets | -3.606E-03 | 0.031 | 0.996 | 0.993–0.999 |
| Constant | -3.013 | 0.037 |  |  |

### Regression model for baseline and demographic data

We combined the parameters patient age and the initially determined platelet count into a self-consistent linear regression model for favorable outcome. Regression coefficients, odds ratios (and associated confidence intervals), and p-values with respect to the model are shown in Table 2.

ROC analysis gives a value for the area under the curve of 0.715 for this model; ROC curve is shown in Fig 1. Setting the separation value in the regression model to 0.5 (probability of survival and death is equal in the regression model, which means that the linear combination of the variables is zero) corresponds to the point marked with a blue arrow in Fig 1: here, the sensitivity of the model is at 44.7, the specificity at 91.4. This point was chosen because the slope of the curve changes here. With this separation value, the linear regression model can be simplified, and a cut-off for the linear combination of the parameters age and platelet count

$$t_{ini} = age - \frac{platelets}{17.65}$$

can be determined from the regression coefficients: For $t_{ini}<47.4$ (78% of the patients in our study fall into this range), mortality for the cohort considered in this study is 26.0%. For $t_{ini}>47.4$ (22% of the patients in our study fall into this range), mortality is 75.0%.

### Basic data of VV ECMO

The period from symptom onset until the start of VV ECMO was 14.6±6.6 days for survivors and 16.3±6.7 days for non-survivors (p = 0.137). Survivors had a shorter period between admission to the ICU and the start of VV ECMO (7.1±6.0 vs. 10.0±6.4 days, p = 0.008) and between intubation and the start of VV ECMO (4.7±5.4 vs. 7.1±6.4 days, p = 0.016) than non-survivors. VV ECMO had been started at an external hospital in 57 (70.4%) survivors and in 29 (60.4%) non-survivors (p = 0.167). Survivors had received VV ECMO for 29.1±20.8 days and non-survivors for 36.8±28.1 days (p = 0.088).

### Course of ICU treatment within the first 10 days after the start of VV ECMO

Examination of the course of parameters of blood diagnostics, dosage of catecholamines, circulatory parameters, VV ECMO, ventilation therapy, and ICU scores (in sum 65 parameters) yielded a striking result: On day 1 after the start of VV ECMO, survivors and non-survivors showed a significant or even highly significant difference in 8 parameters, on day 3 in 12 parameters, on day 5 in 28 parameters, and on day 10 even in 36 parameters. Thus, survivors and non-survivors showed increasingly pronounced differences over the course of ICU treatment.

Table 3 summarizes the values for these parameters of survivors and non-survivors on day 10 after the start of VV ECMO. The values for all days are provided in 'S2 Table'.

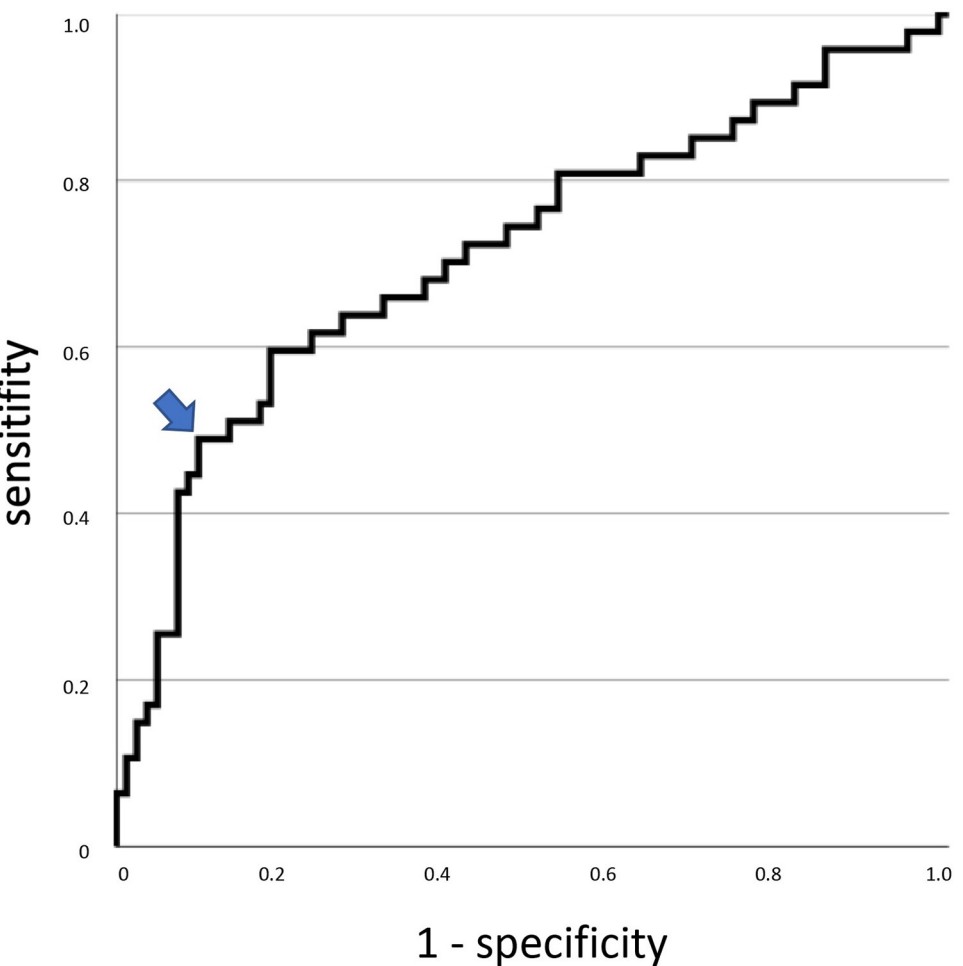

**Fig 1. The receiver operating characteristics curve for linear regression model only including the parameters age and initial platelet count.** The blue arrow marks the point that corresponds to a sensitivity at 44.7 and a specificity at 91.4. Using this point as a separator to divide the patient population into two groups, mortality in the first group is 26% (including 78% of the patients) and 75% in the second group (including 22% of the patients).

Significant differences between survivors and non-survivors are highlighted in orange, highly significant differences in purple. Av, average value for the day; min, minimum value for the day; max, maximum value for the day; $HCO_3^-$, standard bicarbonate; BE, base excess, $Cl^-$, chloride; $paO_2$, partial pressure of oxygen; $paCO_2$, partial pressure of carbon dioxide; Hb, hemoglobin; GFR, glomerular filtration rate; AST, aspartate transaminase; ALT, alanine transaminase; LDH, lactate dehydrogenase; CRP, C-reactive protein; PCT, procalcitonin, WBC, white blood cells; IL6, interleukin 6; INR, International Normalized Ratio; D-Dim, d-dimers; MAP, mean arterial pressure; HR, heart rate; flow, pump flow rate; sweep, sweep gas flow rate; $FiO_2$, fraction of inspired oxygen; HV, oxygenation ratio; PEEP, positive endexpiratory pressure; RMV, respiratory minute volume; TV, tidal volume; Ppeak, peak inspiratory pressure; TISS, Therapeutic Intervention Scoring System; SAPS, Simplified Acute Physiology Score; CI, confidence interval; MD, mean difference. The units of the individual parameters can be found in 'S1 Table'

Box plot diagrams for metric parameters are provided in 'S2 Fig'. Significant differences are marked with an asterisk.

Significant differences in the occurrence of fever between survivors and non-survivors were only found on day 1 (24 vs. 5 patients, p = 0.016). Survivors required renal replacement

**Table 3. Parameters for ICU treatment on day 10 after the start of veno-venous extracorporeal membrane oxygenation (VV ECMO).**

| | Survivors | | | | Non-survivors | | | | 95%-CI of MD | | | p |
|---|---|---|---|---|---|---|---|---|---|---|---|---|
| | mean | SD | min | max | mean | SD | min | max | mean | min | max | |
| pH av | 7.45 | 0.03 | 7.38 | 7.51 | 7.42 | 0.05 | 7.28 | 7.52 | 0.02 | 0.01 | 0.04 | 4.055E-02 |
| pH min | 7.39 | 0.04 | 7.30 | 7.52 | 7.37 | 0.06 | 7.19 | 7.46 | 0.03 | 0.01 | 0.05 | 4.926E-02 |
| pH max | 7.51 | 0.05 | 7.43 | 7.64 | 7.49 | 0.06 | 7.36 | 7.62 | 0.02 | 0.00 | 0.05 | 5.561E-02 |
| $HCO_3^-$ av | 31.7 | 3.7 | 22.3 | 40.6 | 29.3 | 5.0 | 18.0 | 39.1 | 2.3 | 0.6 | 4.1 | 5.374E-03 |
| $HCO_3^-$ min | 29.3 | 3.8 | 20.9 | 39.5 | 26.5 | 5.3 | 13.5 | 37.3 | 2.7 | 0.9 | 4.5 | 2.737E-03 |
| $HCO_3^-$ max | 33.9 | 3.7 | 25.7 | 42.3 | 31.5 | 5.2 | 22.6 | 42.7 | 2.4 | 0.6 | 4.2 | 4.460E-03 |
| BE av | 6.9 | 3.5 | -1.6 | 14.7 | 4.5 | 5.1 | -6.9 | 14.5 | 2.4 | 0.6 | 4.1 | 7.052E-03 |
| BE min | 4.6 | 3.6 | -3.4 | 13.4 | 1.9 | 5.4 | -12.2 | 12.9 | 2.7 | 0.8 | 4.6 | 4.062E-03 |
| BE max | 8.8 | 3.7 | -0.2 | 16.8 | 6.6 | 5.1 | -2.3 | 18.1 | 2.2 | 0.5 | 4.0 | 7.695E-03 |
| $Cl^-$ av | 106.0 | 5.8 | 91.3 | 120.2 | 109.1 | 7.0 | 94.1 | 125.1 | -3.2 | -5.7 | -0.6 | 2.923E-02 |
| $Cl^-$ min | 103.4 | 5.8 | 90.0 | 119.0 | 106.8 | 6.9 | 91.0 | 121.0 | -3.4 | -5.9 | -0.9 | 1.179E-02 |
| $Cl^-$ max | 108.7 | 6.2 | 93.0 | 123.0 | 112.0 | 6.8 | 96.0 | 128.0 | -3.2 | -5.8 | -0.7 | 2.548E-02 |
| $paO_2$ av | 78.0 | 7.5 | 65.4 | 100.9 | 74.2 | 7.9 | 59.3 | 102.1 | 3.8 | 0.8 | 6.8 | 1.310E-02 |
| $paO_2$ min | 66.0 | 8.9 | 33.1 | 85.4 | 61.8 | 8.0 | 37.0 | 73.7 | 4.2 | 1.0 | 7.4 | 1.017E-02 |
| $paO_2$ max | 92.4 | 15.0 | 70.7 | 137.0 | 91.2 | 16.8 | 65.4 | 130.0 | 1.2 | -5.0 | 7.4 | 5.292E-01 |
| $paCO_2$ av | 46.2 | 4.7 | 34.9 | 60.4 | 44.8 | 4.9 | 32.8 | 54.6 | 1.4 | -0.5 | 3.2 | 2.934E-01 |
| $paCO_2$ min | 42.0 | 5.3 | 30.2 | 59.2 | 40.9 | 4.5 | 29.1 | 52.1 | 1.1 | -0.7 | 2.9 | 3.756E-01 |
| $paCO_2$ max | 50.7 | 5.5 | 39.1 | 66.8 | 48.6 | 6.0 | 36.5 | 60.4 | 2.1 | -0.1 | 4.3 | 1.138E-01 |
| Hb av | 9.4 | 1.0 | 8.0 | 12.6 | 9.0 | 0.6 | 8.0 | 9.8 | 0.4 | 0.1 | 0.7 | 6.652E-02 |
| Hb min | 8.7 | 1.1 | 6.6 | 12.0 | 8.3 | 0.7 | 5.4 | 9.5 | 0.4 | 0.1 | 0.7 | 1.032E-01 |
| Hb max | 10.1 | 1.1 | 8.5 | 13.0 | 9.6 | 0.7 | 8.3 | 10.8 | 0.6 | 0.2 | 0.9 | 7.325E-03 |
| Lactate av | 10.7 | 4.2 | 2.7 | 19.7 | 11.7 | 7.9 | 3.7 | 45.6 | -0.9 | -3.5 | 1.7 | 8.482E-01 |
| Lactate max | 14.6 | 6.3 | 3.0 | 36.0 | 16.0 | 12.5 | 5.0 | 71.0 | -1.4 | -5.5 | 2.7 | 7.174E-01 |
| Trop max | 28.2 | 24.7 | 5.7 | 115.0 | 76.3 | 153.1 | 4.5 | 940.0 | -48.2 | -99.6 | 3.3 | 3.532E-03 |
| GFR min | 98.7 | 27.1 | 26.0 | 153.0 | 91.4 | 29.4 | 31.0 | 140.0 | 7.3 | -3.7 | 18.3 | 2.547E-01 |
| Crea max | 0.8 | 0.4 | 0.3 | 3.0 | 0.9 | 0.5 | 0.3 | 2.7 | -0.1 | -0.3 | 0.1 | 7.853E-01 |
| Urea max | 79.3 | 47.1 | 26.0 | 238.0 | 88.3 | 45.5 | 37.0 | 262.0 | -9.1 | -26.8 | 8.6 | 4.599E-02 |
| AST max | 63.8 | 59.0 | 13.0 | 314.0 | 398.4 | 1939 | 17.0 | 12809.0 | -331.7 | -928.6 | 265.3 | 1.829E-03 |
| ALT max | 87.4 | 66.7 | 16.0 | 360.0 | 271.3 | 898.3 | 19.0 | 5952 | -183.9 | -460.8 | 92.9 | 4.571E-02 |
| LDH max | 411 | 152 | 158 | 957 | 782 | 2229 | 236 | 14973 | -372 | -1058 | 315 | 7.041E-01 |
| CRP max | 68.7 | 86.2 | 0.6 | 443.0 | 100.8 | 111.9 | 2.9 | 536.0 | -32.2 | -71.8 | 7.5 | 6.016E-02 |
| PCT max | 0.3 | 0.5 | 0.1 | 3.4 | 1.1 | 2.6 | 0.1 | 14.2 | -0.8 | -1.6 | 0.0 | 3.448E-03 |
| WBC max | 11.5 | 4.8 | 3.8 | 29.9 | 11.1 | 5.7 | 4.0 | 30.0 | 0.4 | -1.7 | 2.5 | 2.672E-01 |
| Ferritin max | 1528 | 2921 | 192 | 19268 | 2153 | 2664 | 295 | 13382 | -625 | -1969 | 719 | 8.752E-02 |
| Lymphos max | 1.3 | 0.6 | 0.3 | 2.8 | 1.4 | 0.8 | 0.3 | 3.7 | 0.0 | -0.3 | 0.2 | 6.119E-01 |
| IL6 max | 116 | 368 | 3 | 2625 | 120 | 213 | 3 | 970 | -4.0 | -115.0 | 107.0 | 5.493E-02 |
| INR av | 1.3 | 0.3 | 0.9 | 2.3 | 1.3 | 0.3 | 1.0 | 2.4 | 0.0 | -0.2 | 0.1 | 4.945E-01 |
| D-Dim. max | 23.9 | 11.6 | 2.4 | 35.0 | 21.3 | 11.0 | 2.5 | 35.0 | 2.6 | -1.8 | 6.9 | 2.488E-01 |
| Platelets av | 199.2 | 88.0 | 53.0 | 463.0 | 147.1 | 57.9 | 47.0 | 242.0 | 52.1 | 25.0 | 79.2 | 1.636E-03 |
| Norepi av | 0.1 | 0.2 | 0.0 | 1.1 | 0.3 | 0.5 | 0.0 | 2.9 | -0.1 | -0.3 | 0.0 | 2.869E-01 |
| Norepi max | 0.2 | 0.3 | 0.0 | 1.5 | 0.5 | 0.8 | 0.0 | 4.0 | -0.2 | -0.5 | 0.0 | 2.633E-01 |
| MAP av | 78.3 | 7.6 | 65.6 | 95.0 | 75.8 | 7.4 | 57.5 | 93.2 | 2.4 | -0.4 | 5.3 | 1.811E-01 |
| MAP min | 63.6 | 7.8 | 34.0 | 80.0 | 60.2 | 7.4 | 45.0 | 80.0 | 3.4 | 0.5 | 6.3 | 1.156E-02 |
| MAP max | 99.6 | 14.5 | 79.0 | 161.0 | 95.5 | 14.5 | 70.0 | 143.0 | 4.1 | -1.4 | 9.3 | 1.184E-01 |
| HR av | 78.1 | 13.6 | 42.6 | 108.1 | 77.7 | 15.1 | 49.6 | 107.3 | 0.4 | -5.2 | 6.0 | 7.614E-01 |
| HR min | 68.8 | 13.8 | 39.0 | 102.0 | 69.6 | 14.2 | 44.0 | 100.0 | -0.8 | -6.2 | 4.6 | 8.594E-01 |

*(Continued)*

**Table 3.** (Continued)

| | Survivors | | | | Non-survivors | | | | 95%-CI of MD | | | p |
|---|---|---|---|---|---|---|---|---|---|---|---|---|
| | mean | SD | min | max | mean | SD | min | max | mean | min | max | |
| HR max | 95.8 | 18.4 | 51.0 | 138.0 | 92.2 | 22.6 | 59.0 | 197.0 | 3.6 | -4.5 | 11.7 | 1.412E-01 |
| flow av | 2.8 | 0.8 | 1.1 | 4.5 | 3.4 | 0.8 | 2.0 | 5.5 | -0.7 | -1.0 | -0.3 | 2.504E-04 |
| flow min | 2.5 | 0.9 | 0.3 | 3.9 | 3.2 | 0.9 | 1.3 | 5.3 | -0.7 | -1.1 | -0.4 | 1.001E-04 |
| flow max | 3.0 | 0.8 | 1.2 | 4.7 | 3.6 | 0.8 | 2.0 | 5.7 | -0.6 | -0.9 | -0.3 | 6.533E-04 |
| sweep av | 5.0 | 2.3 | 0.8 | 12.0 | 6.7 | 2.5 | 2.0 | 11.0 | -1.7 | -2.6 | -0.8 | 4.299E-04 |
| sweep min | 4.7 | 2.5 | 0.0 | 12.0 | 6.3 | 2.4 | 2.0 | 10.0 | -1.6 | -2.6 | -0.7 | 1.048E-03 |
| sweep max | 5.4 | 2.3 | 1.0 | 12.0 | 7.1 | 2.7 | 2.0 | 12.0 | -1.7 | -2.7 | -0.7 | 8.353E-04 |
| FiO2 av | 50.5 | 15.5 | 30.7 | 100.0 | 60.7 | 18.0 | 35.0 | 100.0 | -10.2 | -16.8 | -3.6 | 9.882E-04 |
| FiO2 min | 45.3 | 14.1 | 30.0 | 100.0 | 55.6 | 18.3 | 35.0 | 100.0 | -10.4 | -16.9 | -3.8 | 2.240E-04 |
| FiO2 max | 65.7 | 23.1 | 35.0 | 100.0 | 74.6 | 23.7 | 36.0 | 100.0 | -9.0 | -18.0 | 0.1 | 3.541E-02 |
| HV av | 168.5 | 47.5 | 66.0 | 307.2 | 135.5 | 41.8 | 59.2 | 220.5 | 33.0 | 16.1 | 50.0 | 2.570E-04 |
| HV min | 135.3 | 47.7 | 58.0 | 281.0 | 111.5 | 36.9 | 46.0 | 198.0 | 23.8 | 7.9 | 39.7 | 9.383E-03 |
| HV max | 198.8 | 57.6 | 75.0 | 377.0 | 166.5 | 56.0 | 69.0 | 288.0 | 32.2 | 10.4 | 54.0 | 3.350E-03 |
| PEEP av | 11.6 | 3.1 | 4.8 | 18.4 | 12.5 | 2.8 | 4.8 | 18.9 | -0.9 | -2.0 | 0.3 | 1.202E-01 |
| RMV av | 4.2 | 2.0 | 0.9 | 12.0 | 3.1 | 2.3 | 0.6 | 13.5 | 1.1 | 0.2 | 1.9 | 5.851E-04 |
| VT av | 292.4 | 121.1 | 85.3 | 818.2 | 209.6 | 104.5 | 54.5 | 536.3 | 82.9 | 40.1 | 125.6 | 2.800E-05 |
| Ppeak av | 24.0 | 3.5 | 15.2 | 31.5 | 25.1 | 3.1 | 16.0 | 31.6 | -1.1 | -2.4 | 0.2 | 7.436E-02 |
| TISS | 13.4 | 3.6 | 10.0 | 24.0 | 14.2 | 4.8 | 10.0 | 28.0 | -0.8 | -2.5 | 0.9 | 7.125E-01 |
| SAPS | 22.6 | 6.5 | 11.0 | 47.0 | 28.3 | 9.1 | 16.0 | 55.0 | -5.6 | -8.8 | -2.5 | 3.781E-04 |

therapy less often than non-survivors (day 1: 0 vs. 6 patients, p = 0.002; day 3: 1 vs. 6 patients, p = 0.010; day 5: 2 vs. 7 patients, p = 0.012; day 10: 2 vs. 7 non-survivors, p = 0.013). The two groups did not differ with regard to the administration of unfractionated or low molecular weight heparin, argatroban, or acetylsalicylic acid.

## Regression model for parameters on day 10 after the start of VV ECMO

Because the above-mentioned analysis showed that, on day 10, the differences between survivors and non-survivors were more distinctive than on the earlier days, we chose day 10 to generate a second linear regression model for outcome with the parameters available for that day. By day 10, 5 patients had already died, VV ECMO had already been terminated in 9 patients, and 2 patients had one of the relevant parameters for the model presented below missing for that day. Therefore, the data of 113 patients (out of the initial 129 patients) were used for further analysis. As in the model described above, we used the parameters age and platelet count and combined them with other highly significant parameters. The best parameters for combination were found to be average VV ECMO flow, minimum paO$_2$, and average blood pH on that day. Calculated regression coefficients, significance for the model, and odds ratio with confidence interval are summarized in Table 4. Multicollinearity between the parameters was excluded (S3 Table).

For this model, an ROC analysis was performed (Fig 2), in which the area under the curve has a size of 0.908. Thus, from these parameters, the following value t$_{10}$ can be calculated by linear combination:

$$t_{10} = 218.069 + 173.228 * 10^{-3} * age - 165.850 * 10^{-4} * Platlets\ av + +116.608 * 10^{-2} * Flow\ av - 101.310 * 10^{-3} * PaO_2\ min - 299.329 * 10^{-1} * pH\ av$$

**Table 4. Regression coefficients, significance, and odds ratio with confidence interval for the regression model calculated with the parameters age, average platelet count on day 10 (Platelets av), average pump flow rate on day 10 (Flow av), minimum partial pressure of oxygen on day 10 (paO$_2$ min), and average pH on day 10, multiplied by 100 (pH av*100).**

| | Regression coefficient | p-value | Odds ratio | Confidence interval (95%) for odds ratio |
|---|---|---|---|---|
| Age | 173.228E-03 | <0.001 | 1.189 | 1.090–1.297 |
| Platelets av | -165.850E-04 | <0.001 | 0.984 | 0.975–0.992 |
| Flow av | 116.608E-02 | 0.003 | 3.209 | 1.502–6.856 |
| PaO$_2$ min | -101.310E-03 | 0.005 | 0.904 | 0.842–0.970 |
| pH av*100 | -299.329E-03 | <0.001 | 0.741 | 0.625–0.879 |
| Constant | 218.069 | <0.001 | | |

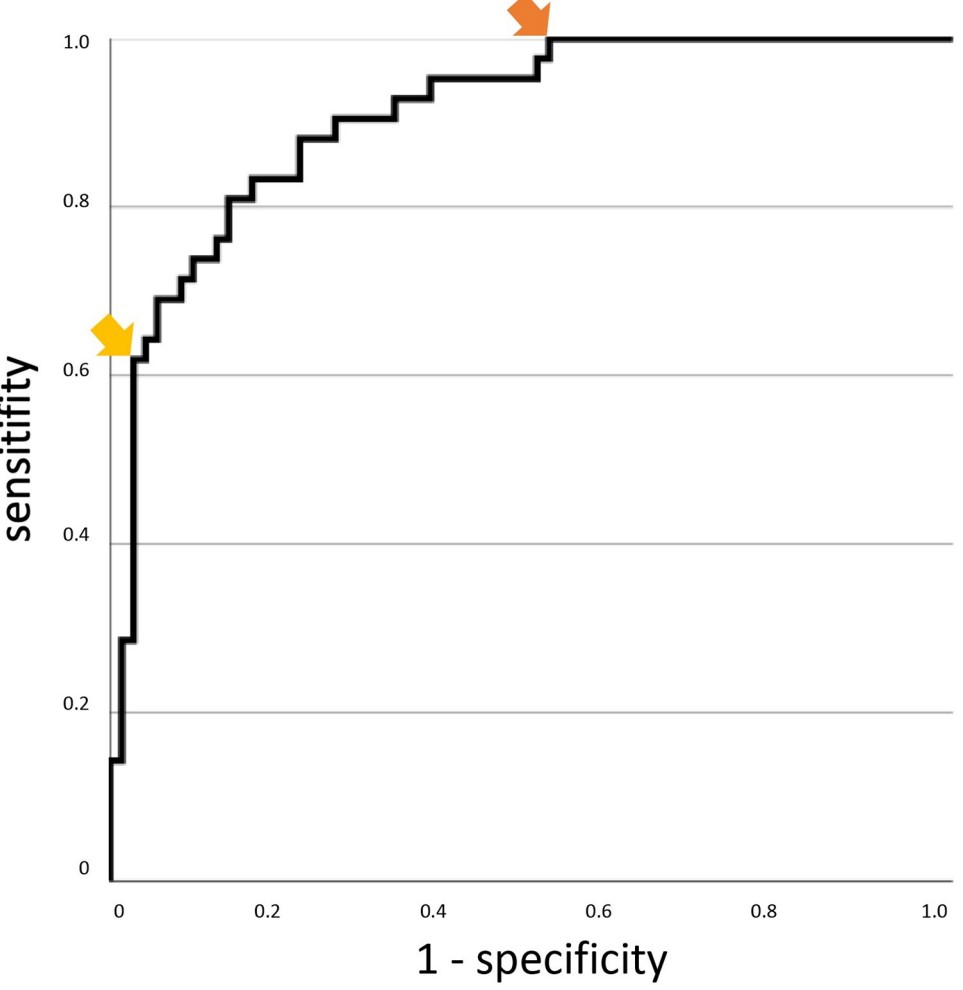

**Fig 2. ROC curve for linear regression model calculated with the parameters age, average platelet count on day 10, average pump flow rate on day 10, minimum partial pressure of oxygen on day 10, and average pH on day 10 after the start of veno-venous extracorporeal membrane oxygenation.** The yellow arrow is at a separation value of 0.85 with a sensitivity of 61.9 and a specificity of 97.2. Of the patient cohort considered in this study, 28 patients (25%) were above this cut-off, 26 non-survivors and 2 survivors (mortality 93%). The orange arrow marks the separation value -2.5 with a sensitivity of 100.0 and a specificity of 47.9. 34 (30%) patients in this study were below the cut-off value of -2.5, all of them survivors (mortality 0%). The group between these two cut-offs comprises 45% of the patients with a mortality rate of 31%.

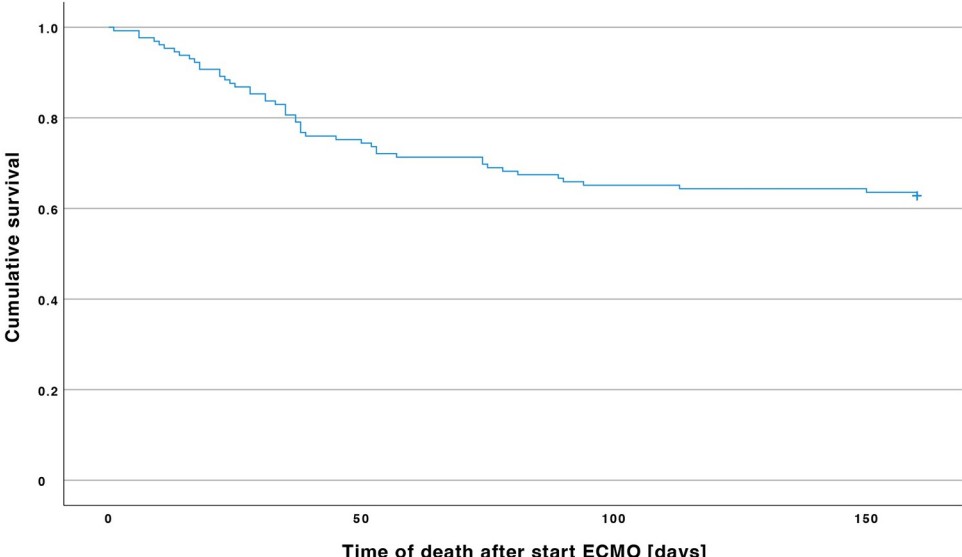

**Fig 3. Kaplan-Meier estimator.** Out of 129 patients treated with veno-venous extracorporeal membrane oxygenation (VV ECMO), 48 (37.2%) patients had died in the intensive care unit. In the group of non-survivors, the last patient had died 134 days after the start of VV ECMO.

Using the ROC curve, two cut-off values can be determined for $t_{10}$ (yellow and orange arrows in Fig 2): If $t_{10} > 0.85$, the probability of survival is very low. In the patients considered in our study, $t_{10}$ was greater than this cut-off in 28 of the 113 patients still included on day 10 (25%), and only two 2 of them had survived (mortality 93%). If $t_{10} < -2.5$ (which was the case in 34 of 113, i.e. 30% of the patients), the probability of survival is very high (100% of the patients in our study, mortality 0%). In the group between these two cut-offs (45% of the patients), the mortality rate is 31%.

### Description of fatal cases

Mean time of death after the start of VV ECMO was 43.0±32.0 days (Fig 3). 39 patients (81.3% of non-survivors) had died while receiving VV ECMO. The cause of death was multiorgan failure in 21 (43.8%) patients, respiratory failure in 17 (35.4%) patients, cerebral bleeding in 5 (10.3%) patients, fatal bleeding in 2 (4.2%) patients, cardiac failure in 1 (2.1%) patient, cerebral ischemia in 1 (2.1%) patient, and intestinal ischemia 1 (2.1%) patient.

### Discussion

Survival data of patients receiving VV ECMO for acute respiratory failure due to COVID-19 should be similar to survival data of patients receiving VV ECMO for other causes of acute respiratory distress syndrome (ARDS) [6]. This study included the data of 129 patients, of whom 81 had survived (mortality 37.2%). Although the mortality rates of patients receiving VV ECMO for acute respiratory failure due to COVID-19 have been reported higher in Germany than in other countries, our rates are still in the internationally reported range [2–4].

The current Extracorporeal Life Support Organization (ELSO) guideline on ECMO in patients with COVID-19 lists several relative and absolute contraindications. Yet, these recommendations are based on the data of ICU patients with conventionally treated COVID 19 and on already existing VV ECMO risk prediction models derived from ICU patients without COVID-19 (6). Hard data to predict outcome and thus to select patients who are most likely to

benefit from VV ECMO are still rare. The decision for or against VV ECMO has often to be made quickly; therefore, it should be based on parameters that are easy to collect. The reasons for discontinuing VV ECMO in patients with COVID-19 cited in the ELSO guideline are severe neurological insult, no heart or lung recovery with no possibility of a durable device implantation or transplant, and progressive multiple organ failure despite timely and optimal cardiopulmonary support. These basically abstract statements, however, can in turn be assessed differently by the treatment team for each individual patient. Specific cut-off values for individual parameters will certainly not be the sole basis for the decision to terminate ongoing VV ECMO but may be an important factor for facilitating the decision-making process in the context of perceived clinical evolution.

The set of parameters available for most of our patients just before the start of VV ECMO was relatively small, because 86 (66.7%) patients had already undergone implantation of the EMCO cannulas and start of VV ECMO in an external hospital. The procedures were carried out by a mobile team of the University Medical Center Regensburg, and the patients were subsequently transported to our hospital by ambulance or helicopter. The documentation of laboratory diagnostic values, circulatory monitoring, and ventilator parameters of the different external hospitals showed sometimes significant differences.

Significance analyses identified two parameters from the set of baseline and demographic data that were associated with higher mortality, namely age and platelet count. The association between advanced age and more severe courses of disease as well as a higher risk of death from COVID-19 has been repeatedly reported [7, 8]. Similarly, the risk of death is known to increase with age in patients undergoing VV ECMO because of respiratory failure, independent of the presence of COVID-19 [9, 10]. For patients with COVID-19 who receive VV ECMO recently published studies had also revealed an association between advanced age and increased mortality [11–14].

Analysis of the data collected on day 1, 3, 5, and 10 after the start of VV ECMO indicated an increasing divergence between the two groups of patients. This divergence is evidenced by an increasing number of parameters with significant differences between survivors and non-survivors from time point to time point. In addition, the p-values of most parameters that significantly differ on day 10 became smaller over time, indicating an increasing difference between the groups. Examples are the parameters SAPS, RMV average, and $SpO_2$ minimum, but also VV ECMO flow rate and sweep gas flow.

Our study yielded a highly significant difference in the mean platelet count between survivors and non-survivors, not only before but also throughout the observation period. Two effects are likely to play a role here: platelet activation and consumption are associated with poorer prognosis [15], which is particularly true for SARS-CoV-2 viruses [16]. This effect triggered by viruses or in a similar form by bacteria has also been described in detail elsewhere [17]. A second effect is discussed in that work, namely how an additional decrease in the platelet count occurs during VV ECMO. This point was also examined elsewhere [18] and is so far only partially understood. Low platelet count is known to be often correlated with bleeding events [19] and intracerebral hemorrhage [20] during VV ECMO, as studied in patients without COVID-19. Consistent with our findings, a recently published study of COVID-19 patients treated with VV ECMO found significantly lower platelet counts in non-survivors, however, only 32 patients were included in this study [21].

The aim of this work was not only to identify the relevance of individual parameters for predicting outcome but ideally to generate models that are more informative than individual parameters by combining such parameters. A simple linear regression model of age and platelet count before the start of VV ECMO enabled us to allocate patients into two groups by choosing a cut-off value for the calculated linear parameter combination, in which mortality

was 26% in one group and 75% in the other. Considering that neither pre-existing diseases nor other parameters describing a patient's health status are included in this model, this simple model shows surprisingly good predictive power.

The model for day 10 is also a simple linear regression model but includes five parameters. The parameters are hardly correlated with each other, which is also shown by the fact that every single parameter is significant with respect to the model; this way, a mathematically meaningful model was created. ROC analysis showed a special shape of the curve: a shift of the separation value created a curve in which sensitivity behaved optimally in one section and specificity in the other. Therefore, two cut-off values were derived, which divided the patients into three groups: a group, in which mortality was 0%, a group in which mortality was almost 100%, and a third group with values in between.

The model for day 10 includes the parameter average blood pH in addition to the already discussed parameters age and platelet count. The average blood pH value should be viewed as a parameter that can be used to assess the clinical development in ICU patients with COVID-19. In previous studies and in line with data published by Choron et al. [22], we were able to identify blood pH to be of high importance for predicting unfavorable outcome in patients with COVID-19 and ARDS who require ICU therapy [23, 24].

Finally, $paO_2$ and VV ECMO pump flow on day 10 were included in the model. The extent of pulmonary oxygenation impairment is directly reflected by $paO_2$ and indirectly reflected by VV ECMO pump flow. Patients with severe, persistently impaired pulmonary oxygenation despite 10 days of VV ECMO are likely to have more severe lung injury, which also should influence the expected outcome. At least it could be shown that patients with COVID-19 with persistent severe restrictive lung dysfunction and low compliance have increased mortality rates [25].

## Limitations

This is a monocentric study with retrospective data collection. The fact that only 129 patients could be included limits the validity of the study. Furthermore, patients with different mutation variants of SARS-CoV-2 were included, and we now know that the clinical courses differed depending on the underlying mutation variant of SARS-CoV-2.

## Conclusions

Before initiation of ECMO therapy, subsequent survivors and non-survivors differed primarily in age and platelet count. A linear regression model could be calculated. By entering the patient's age and platelet count into this formula, a numerical value is obtained that allows estimation of the prognosis in terms of survival or death at ICU. This may provide guidance in deciding whether to begin VV ECMO therapy.

At 10 days after initiation of VV ECMO therapy, there were highly significant differences in age, platelet count, average blood pH, minimum paO2, and average pump flow in the later survivors and non-survivors. Again, a linear regression model could be calculated and cutoff values could be derived to estimate the prognosis with high probability. If there is any doubt as to whether it is appropriate to continue intensive care therapy, this can provide valuable additional information.

Because of the limited number of patients, however, the models need to be tested in a larger patient collective.

## Supporting information

**S1 Appendix. Underlying data for all patients.**
(XLSX)

**S2 Appendix. Technical data for ECCMO therapy.**
(XLSX)

**S1 Table. Complete list of parameters.**
(DOCX)

**S2 Table. Listing and comparison of survivors and non-survivors' values for all parameters collected on days 1, 3, 5, and 10 after the start of VV ECMO.**
(DOCX)

**S3 Table. Calculation of correlation coefficients (Pearson) for the parameters of the regression models.**
(DOCX)

**S1 Fig. Flow-chart patient inclusion.**
(PDF)

**S2 Fig. Box-plot diagrams of metric data.**
(PDF)

## Author Contributions

**Conceptualization:** Bärbel Kieninger, Maik Foltan, Martin Kieninger.

**Formal analysis:** Bärbel Kieninger, Magdalena Kilger, Martin Kieninger.

**Investigation:** Bärbel Kieninger, Maik Foltan, Martin Kieninger.

**Methodology:** Bärbel Kieninger, Martin Kieninger.

**Project administration:** Martin Kieninger.

**Software:** Bärbel Kieninger.

**Supervision:** Martin Kieninger.

**Validation:** Michael Gruber, Bernd Salzberger.

**Visualization:** Bärbel Kieninger, Magdalena Kilger, Martin Kieninger.

**Writing – original draft:** Bärbel Kieninger, Martin Kieninger.

**Writing – review & editing:** Maik Foltan, Michael Gruber, Dirk Lunz, Thomas Dienemann, Stephan Schmid, Bernhard Graf, Clemens Wiest, Matthias Lubnow, Thomas Müller, Bernd Salzberger, Wulf Schneider-Brachert.

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
