## [Decision Letter · Decision Letter 0]

27 Dec 2022

PONE-D-22-22958Prognostic factors for favorable outcomes after veno-venous extracorporeal membrane oxygenation in critical care patients with COVID-19PLOS ONE

Dear Dr. Kieninger,

Thank you for submitting your manuscript to PLOS ONE. After careful consideration, we feel that it has merit but does not fully meet PLOS ONE’s publication criteria as it currently stands. Therefore, we invite you to submit a revised version of the manuscript that addresses the points raised during the review process.

We look forward to receiving your revised manuscript.

Kind regards,

Lakshmi Kannan

Academic Editor

PLOS ONE

Journal Requirements:

Martin Kieninger:

I have read the journal's policy and the authors of this manuscript have the following competing interests: I currently serve as an Academic Editor for PLoS ONE.

Reviewers' comments:

Reviewer's Responses to Questions

**Comments to the Author**

1. Is the manuscript technically sound, and do the data support the conclusions?

Reviewer #1: Yes

Reviewer #2: Yes

2. Has the statistical analysis been performed appropriately and rigorously? 

Reviewer #1: Yes

Reviewer #2: Yes

3. Have the authors made all data underlying the findings in their manuscript fully available?

Reviewer #1: Yes

Reviewer #2: Yes

4. Is the manuscript presented in an intelligible fashion and written in standard English?

Reviewer #1: Yes

Reviewer #2: Yes

5. Review Comments to the Author

Reviewer #1: the authors present a review of characteristics associated with survival in VV ECMO Covid patients.

1. was the cutoff for ECMO initiation at 14 d of mech vent due to your center's criteria used for all patients or just COVID ?

2. Was there also info on survival to discharge not just ICU?

While the association of plt count and outcome is interesting, one wonders if this is clinically significant. There is also no data on how many plt transfusion were required or given between groups, the number of circuit changes etc, all of whihc can affect plat ct.

3. It seems that the NS had more signs of infection and being sicker overall at day 10--higher flows, need for higher Fio2, higher PCt, higher lactate.

Reviewer #2: #### This is a retrospective observational study of patients placed on VVECMO for ARDS

#### Comments

1. important work to determine the factors that most correlate with poor outcomes among patients placed on VVECMO. Work needs to be done to solidify eligibility criteria for VVECMO - especially in COVID-19 patients. The fact that age and platelet count were significant is helpful information.

2. The manuscript is a little complicated for me (as a clinician) - I would recommend review by a biostatistician.

3. I would encourage the investigators to simplify the take home message in the discussion and how that take home message fits into the existing literature. For instance, if age and platelet count are associated with the worst outcomes, is this finding in-line with existing research (e.g. is the age range similar?).

4. I could use some clarification of how the "cut off" values were determined - if Youden's index? (Or something similar?)

6. PLOS authors have the option to publish the peer review history of their article (what does this mean?). If published, this will include your full peer review and any attached files.

Reviewer #1: No

Reviewer #2: No

---

## [Author Response · Author response to Decision Letter 0]

29 Dec 2022

Reviewer #1:

1. Was the cutoff for ECMO initiation at 14 d of mech vent due to your center's criteria used for all patients or just COVID?

At our center, we do not have a fixed cutoff until when VV ECMO therapy is started after initiation of ventilator therapy. In justified cases, the implantation of ECMO can still be performed a few weeks after the start of ventilation therapy as a case-by-case decision. This was also done in COVID-19 patients.

In the present study, we were interested in capturing the initial phase of COVID-19 disease with severe respiratory failure and ECMO therapy. The cutoff of 14 days was chosen because, of the 131 patients included, 129 were started on VV ECMO during this period; only 2 patients were implanted on VV ECMO later (see S1 Figure).

2. Was there also info on survival to discharge not just ICU?

Unfortunately, we have data on this for only a small fraction of patients, as most patients were transferred from the ICU to either a rehabilitation facility or a non-tertiary hospital.

While the association of plt count and outcome is interesting, one wonders if this is clinically significant. There is also no data on how many plt transfusion were required or given between groups, the number of circuit changes etc, all of which can affect plat ct.

The aim of our study was to investigate a variety of parameters relevant to intensive care medicine with regard to their suitability as prognostic markers. Platelet count was identified as a very important prognostic parameter in COVID-19 patients with VV ECMO therapy. However, due to the design of our study, no further information on the cause of this and the actual clinical relevance can be derived. As a next step, the identified prognostically relevant parameters would now have to be further investigated in a targeted manner.

However, we have attached another file as a supplement in which technical data on ECMO therapy can be found (S3 Appendix). It also lists the number of platelet concentrate transfusions during the entire ICU stay and the number of membrane oxygenators required during the entire period of ECMO therapy for each patient.

The number of patients who required transfusion of at least one platelet concentrate during the ICU stay did not differ between survivors and non-survivors (23.4% vs. 22.9%).

3. It seems that the NS had more signs of infection and being sicker overall at day 10--higher flows, need for higher Fio2, higher PCt, higher lactate.

This is consistent with our observations. While on the first days after start of ECMO therapy the differences between the survivors and the non-survivors are far from clear, a clear difference is already measurable on day 10, although the event of death is still several weeks away in a large proportion of the non-survivors (see Figure 3). This is the reason why we used day 10 to calculate our model. Conversely, at the time start of ECMO therapy, the patients did not differ measurably with respect to the parameters typically reflecting disease severity.

Reviewer #2:

1. Important work to determine the factors that most correlate with poor outcomes among patients placed on VVECMO. Work needs to be done to solidify eligibility criteria for VVECMO - especially in COVID-19 patients. The fact that age and platelet count were significant is helpful information.

Thank you for this feedback. The intention for initiating this study was to generate further data to make meaningful decisions about which COVID-19 patients should receive therapy with VV ECMO.

2. The manuscript is a little complicated for me (as a clinician) – I would recommend review by a biostatistician.

The team of authors consists of experienced intensive care physicians, but also two scientists with appropriate mathematical training (Bärbel Kieninger, Bernd Salzberger). Bärbel Kieninger was responsible for the statistical calculations and developed the mathematical models presented in the manuscript.

3. I would encourage the investigators to simplify the take home message in the discussion and how that take home message fits into the existing literature. For instance, if age and platelet count are associated with the worst outcomes, is this finding in-line with existing research (e.g. is the age range similar?).

Our manuscript was originally submitted in August 2022. In the meantime, further studies on ECMO therapy in COVID-19 patients have been published. We performed a new literature search and found in addition to the study published by Bergman et al. three further papers that also showed a correlation between older age and higher probability of death. The mean age of patients included in these studies was basically in a similar range to the age of the patients in our study (our study: survivors 52.6 year, non-survivors 57.4 years; Bergman et al.: survivors 49.1 years, non-survivors 55.5 years; Joshi et al: survivors 46.5 years, non-survivors 51.8 years; Raff et al.: survivors 40.6 years, non-survivors 53.1 years; Hettlich et al.: survivors 47.9 years, non-survivors 56.7 years).

In addition, a study was published in the meantime in which significantly lower platelet counts were measured in COVID-19 patients with ECMO therapy in agreement with our results in non-survivors.

We have included the results of these studies in the discussion (line 332-333, line 350-352) and added the corresponding sources in the references.

We have tried to make the take home message in the Conclusion easier to understand and have reformulated it accordingly (line 391-403):

Before initiation of ECMO therapy, subsequent survivors and non-survivors differed primarily in age and platelet count. A linear regression model could be calculated. By entering the patient's age and platelet count into this formula, a numerical value is obtained that allows estimation of the prognosis in terms of survival or death at ICU. This may provide guidance in deciding whether to begin VV ECMO therapy.

At 10 days after initiation of VV ECMO therapy, there were highly significant differences in age, platelet count, average blood pH, minimum paO2, and average pump flow in the later survivors and non-survivors. Again, a linear regression model could be calculated and cutoff values could be derived to estimate the prognosis with high probability. If there is any doubt as to whether it is appropriate to continue intensive care therapy, this can provide valuable additional information.

Because of the limited number of patients, however, the models need to be tested in a larger patient collective.

4. I could use some clarification of how the "cut off" values were determined - if Youden's index? (Or something similar?)

For the calculation of the first model (model from the initial known parameters before VV ECMO), the value that mathematically best separated the 2 groups was chosen as cutoff, which corresponds to the maximization of the Youden index in the present data set.

The ROC curve in the second model (model at day 10) shows a peculiarity in that it first runs very close to the y-axis (very high specificity) and then in the later course runs parallel to the x-axis at a sensitivity of 1. Because of this shape, it is convenient to define 2 cutoffs, each of which keeps one of the classification errors false-positive and false-negative low, but at the same time satisfies the condition that the resulting subgroups must also contain a significant proportion of the patient cohort. The two cutoffs result in three patient groups, one with very high mortality and the second with very low mortality. For these two groups, the model is able to estimate the outcome in our cohort with high probability. However, for the third resulting group, no reasonable estimate of the outcome is possible. This method with two cutoffs seems to be much more helpful in clinical practice than if only one cutoff had been generated by optimizing both classification errors simultaneously.

---

## [Editor Report · Decision Letter 1]

3 Jan 2023

Prognostic factors for favorable outcomes after veno-venous extracorporeal membrane oxygenation in critical care patients with COVID-19

PONE-D-22-22958R1

Dear Dr. Kieninger,

We’re pleased to inform you that your manuscript has been judged scientifically suitable for publication and will be formally accepted for publication once it meets all outstanding technical requirements.

Kind regards,

Lakshmi Kannan

Academic Editor

PLOS ONE
---

## [Editor Report · Acceptance letter]

9 Jan 2023

PONE-D-22-22958R1 

Prognostic factors for favorable outcomes after veno-venous extracorporeal membrane oxygenation in critical care patients with COVID-19 

Dear Dr. Kieninger:

I'm pleased to inform you that your manuscript has been deemed suitable for publication in PLOS ONE. Congratulations! Your manuscript is now with our production department. 

Kind regards, 

on behalf of

Dr. Lakshmi Kannan 

Academic Editor

PLOS ONE